# *We choose*: Adolescent girls and young women's choice for an HIV prevention product in a cross-over randomized clinical trial conducted in South Africa, Uganda, and Zimbabwe

**Millicent Atujuna**[1]*, Kristin Williams[2], Sarah T. Roberts[2], Alinda Young[2], Erica N. Browne[2], Nomvuyo T. Mangxilana[1], Siyanda Tenza[3], Mary Kate Shapley-Quinn[2], Thelma Tauya[4], Kenneth Ngure[5,6], Ariane van der Straten[7,8]

1 Desmond Tutu HIV Centre, Institute of Infectious Disease and Molecular Medicine, Faculty of Health Sciences, Cape Town, South Africa, 2 RTI International, Research Triangle Park, NC, United States of America, 3 WITS Reproductive Health and HIV Institute (WRHI), Johannesburg, South Africa, 4 University of Zimbabwe Clinical Trials Research Centre (UZ-CRC), Harare, Zimbabwe, 5 School of Public Health, Jomo Kenyatta University of Agriculture and Technology, Nairobi, Kenya, 6 Department of Global Health, University of Washington, Seattle, WA, United States of America, 7 Dept of Medicine, Center for AIDS Prevention Studies, University of California, San Francisco, San Francisco, CA, United States of America, 8 ASTRA consulting, Kensington, CA, United States of America

* Millicent.atujuna@hiv-research.org.za

**Data Availability Statement:** All relevant data are within the paper in tables. Additional study data are

## Abstract

With new pre-exposure prophylaxis (PrEP) modalities for HIV prevention becoming available, understanding how adolescent girls and young women (AGYW) navigate through PrEP options is essential, including factors underlying their choice. Through 16 focus group discussions (FGDs) and 52 in-depth interviews (IDIs) from REACH, an open-label crossover study in which AGYW were allocated 1:1 (between 06 February 2019 and 18 March 2020) to receive oral PrEP for six months and the dapivirine ring for six months, in a randomized sequence, followed by a 6-month period where either product (or neither) could be chosen, we explored decision-making process and product choice, using a mixed inductive-deductive analytical approach. Key themes included the desire to remain HIV-negative and weighing product attributes through experiential learning. Product triability appeared important in informing product choice as individual circumstances changed or assuaging side effects with a given product. Approved biomedical prevention innovations may also benefit from hands-on experience to help with adoption and use during real-world implementation. Furthermore, support from trusted providers will remain critical as AGYW contemplate navigating through PrEP options and choice.

## Introduction

Considerable progress has been made in the development of new HIV prevention technologies including, oral pre-exposure prophylaxis (PrEP) [1, 2], the monthly dapivirine vaginal ring

available upon request from the Microbicide Trials Network by submission of a Dataset Request Form available at http://www.mtnstopshiv.org/resources. Interested parties would be able to access these data in the same manner as the authors. The authors did not have any special access privileges that others would not have.

**Funding:** This study was designed and implemented by the Microbicide Trials Network (MTN) funded by the National Institute of Allergy and Infectious Diseases through individual grants (UM1AI068633, UM1AI068615 and UM1AI106707), with cofounding from the Eunice Kennedy Shriver National Institute of Child Health and Human Development and the National Institute of Mental Health, all components of the U.S. National Institutes of Health (NIH) The funder was had no role in study design, data collection and analysis, decision to publish, or preparation of the manuscript.

**Competing interests:** The authors have declared that no competing interests exist.

(ring) [3, 4], and, most recently, long-acting injectable cabotegravir (CAB-LA) [5, 6]. The World Health Organization (WHO) has recommended the use of oral PrEP, the ring, and CAB LA for HIV prevention for women at substantial risk of HIV infection as part of combination prevention approaches [7, 8]. With various biomedical HIV prevention methods becoming available to those who need it, including cisgender adolescent girls and young women (AGYW) [9–11], it is important to characterize how individuals will navigate choice among these options, including understanding the underlying factors for choice.

HPTN 082, 3P, and POWER studies [12–14] have assessed how AGYW responded to oral PrEP, investigating various PrEP delivery models and support strategies to facilitate uptake, implementation, and persistence. In qualitative studies nested within clinical trials, side effects, stigma, and certain product characteristics (e.g., frequency of dosing, conspicuousness) were reported as barriers to PrEP adherence, while perceived HIV risk, social support, product efficacy, and other attributes (e.g., dosage form familiarity) were reported as facilitators [15–17]. Previous acceptability and preference studies of various delivery forms for prevention, using placebo products, found that discreetness and longer duration of protection to minimize user burden were favored, along with products that did not interfere with intimate relationships and provided HIV prevention for unanticipated situations [18]. Furthermore, previous experience with long-acting contraception (e.g., implants) influenced the choice of product [19], and preference also varied by geographical location [20]. However, research examining product preference and choice using products with active HIV prevention drugs is still lacking, and this is the first of its kind testing preference for oral PrEP containing Emtricitabine/Tenofovir Disoproxil Fumarate and the vaginal ring containing Dapivrine, by offering participants the option to experience both products before choosing their preferred HIV prevention product'.

We explored product choice among AGYW who participated in the Microbicide Trials Network (MTN 034) Reversing the Epidemic in Africa with Choices in HIV Prevention (REACH) trial to understand what drives individuals to choose certain prevention options over others, what shapes their decision-making processes, and how they navigate through the choice process. We specifically aimed to understand how experiencing a product may inform choice and future rollout of multiple prevention options using the REACH data.

## Methods and materials

### Study, design and setting

REACH was a Phase 2a, randomized, open-label crossover trial conducted among 247 HIV-negative AGYW (16-21-year-old) assumed to have been assigned female at birth in South Africa (Cape Town and Johannesburg), Zimbabwe (Harare) and Uganda (Kampala) between 06 February 2019 and 18 March 2020. The main objective was to assess the safety of and adherence to the ring and pill among AGYW and to understand product preference between the two products (21). The trial had 3 study periods of 6 months each. In the first period, participants were randomly assigned to use the ring or pill for six months, after which they were switched to using the other product for another six months (also referred to as the crossover period). In the third (or choice) period, participants could choose to use either product -or neither- for six months. While the REACH study design and participant eligibility are published elsewhere [21–23], we draw the reader's attention to the qualitative study embedded within the REACH trial from which data this paper is drawn. Specifically, for this analysis on product choice, we used the Focus Group Discussions (FGDs) and In-depth Interviews(IDIs) data conducted during the choice period, which sought to identify factors influencing the choice between ring, pill, or neither product from a purposively selected sub-sample of AGYW participating in the trial. Importantly, participants could switch between study products or neither

| Signal* | Interpretation of product use results during the crossover at month 6 | |
|---|---|---|
| | **Tablets** | **Ring** |
|  | High use (> 700 fmol/punch)' | High release (rate > 4.0mg/month) |
|  | Some use (16.6 - 699 fmol/punch' | Some release (0.9mg/month < rate <= 4.0mg/mo |
|  | No use (< 16.6 fmol/punch) | 'No use (rate <= 0.9mg/month) |

**Fig 1. Drug level feedback during crossover periods (N = 109).**

during the *choice period*. Furthermore, the study employed several adherence-support strategies, including participant-centered counseling and peer adherence support [24]. Drug level feedback results were provided to participants based on intracellular tenofovir-diphosphate (TFV-DP) levels in red blood cells (for oral PrEP) and residual dapivirine (DPV) levels in the ring (Fig 1). These results were conveyed to the participants through a visual tool and color coding scheme[14].

## Qualitative study sample

From a total of 247 AGYW randomized in REACH, we planned to enroll a sub-set sample of 144 participants (58.50% of the total sample) for participation in either FGDs or IDIs at four sites (n = 36 per site). In determining the sampling technique and sample size, we employed approaches used in other qualitative studies [25] as well as approaches utilized by our team in previous studies [26, 27], thereby selecting participants stratified by age groups, arms, balanced by arm and site. Our sample size was advised by 'information power'[25] where our recruitment reflected the specificity in areas of inquiry (e.g., choosing a product, choosing neither, or those experiencing unique challenges) and collecting data at specific periods during the study. Additionally, during data collection, we promptly analysed debriefing reports (DRs) from completed interviews before the conclusion of data collection. These debriefing reports synthesized the main themes emerging from each interview, with interviewers completing them within 4–7 days of conducting the interview. This rapid analysis enabled us to ascertain data saturation. The final qualitative sample consisted of a mix of those participating in Focus Group Discussions (FGDs) and In-Depth Interviews (IDIs), including special-case IDIs (CIDIs), non-acceptor IDIs (NIDIs), and 3 sets of longitudinal interviews (or serial IDIs (SIDIs) conducted with 6 participants per site (N = 24), in period 1, at crossover (period 2) and in the choice period (period 3). A total of 119 of the planned 144 participants were enrolled in the qualitative study. This difference resulted from the reduction in the number of participants (from 6–8 per group to 4–6 per group) taking part in the FGDs as per COVID-19 guidelines (see Fig 2)

For this analysis, we utilize data collected in the choice period with 109 of the 119 qualitative participants (see Fig 2), including 16 FGD (4 FGDs per country) collected close to study

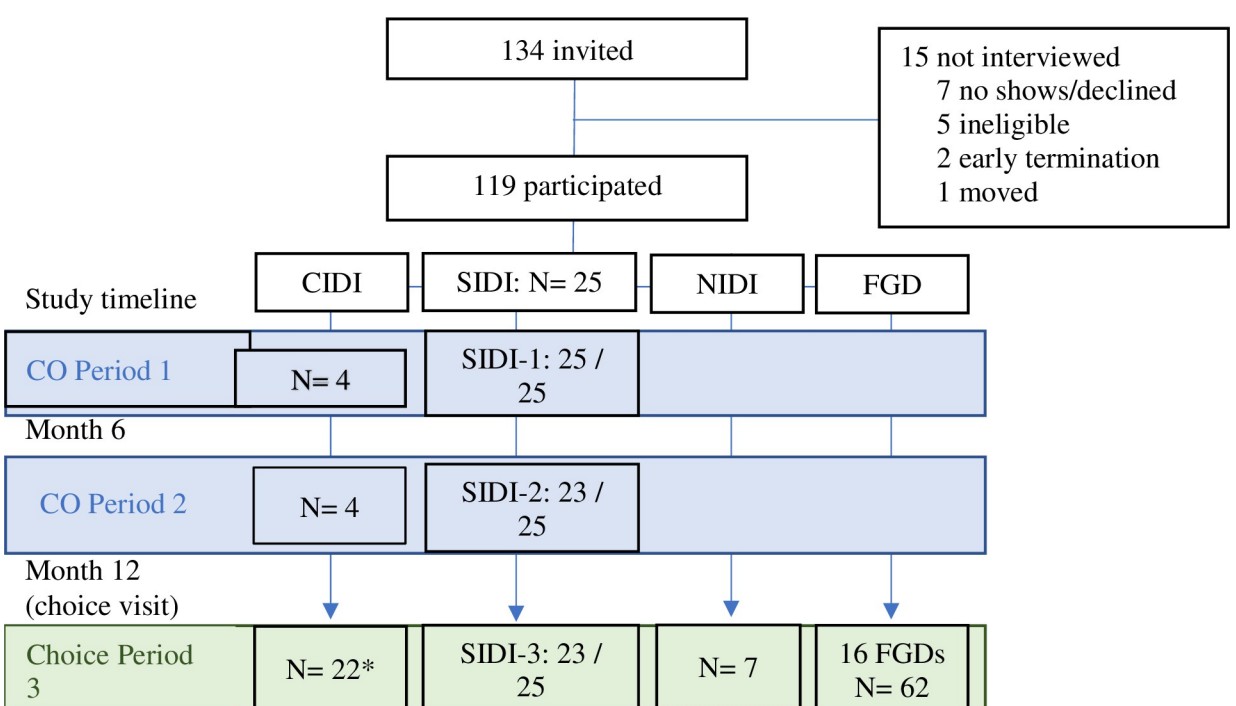

**Fig 2. Consort diagram of completed qualitative activities.** The analysis included IDIs and FGDs from the choice period (period 3).

exit (Month 18) with participant selection based on: i) chosen product; ii) age group (16–18 versus 19–21) and iii) adherence ascertained via drug levels. Additionally, we included the CIDI data of participants who reported experiencing unique situations (e.g., seroconversion, adverse events, social harm, recurring STIs). We further had NIDI data for participants who chose neither product at the choice visit. Finally, we included serial IDI data conducted in the choice period. Our objectives were to explore acceptability and opinions towards the ring and pill to better understand product choice and preferences. These different interviews and FGDs collected and the number of participants in each component are shown in Fig 2 below.

## Procedures for data collection

FGDs and IDIs were conducted in the participant's preferred language (isiXhosa, Zulu (South Africa); Luganda (Uganda); Shona (Zimbabwe) or English) by staff members trained in qualitative research. All interviews and FGDs were conducted in private rooms at the clinical sites by trained in-country interviewers using semi-structured guides [Table 1] exploring product choice/ preference, adherence support, product acceptability, and use experience. SIDIs additionally examined life changes occurring 6 months before the interview. The SIDIs also explored participants' plans for the future. Written informed consent for the qualitative component was obtained as part of the overall informed consent process at enrollment and was verbally reconfirmed before conducting IDIs and FGDs. Interviews and FGDs were audio-recorded, transcribed directly into English locally by one staff member, and reviewed for quality control by a second staff member. Transcripts went through a rigorous quality control and verification process by the Qualitative Data Management Team (QDMT) based in the US, working directly with site staff who collected the data before finalization.

Quantitative data on stated product preference at baseline was assessed for all study participants by Audio Computer-Assisted Self-Interviewing (ACASI). During the choice period,

Table 1. FGD and interview topics for investigation in the choice period.

| FGD guide topics | Discussion points |
|---|---|
| **Product choice/preference**<br>• **Process of decision making**<br>• **What participants think about at choosing** | • What they thought about each product before choosing a product. What were their feelings?<br>• Everyone got to choose between using the ring or pill or choosing neither; what were their thoughts about when making a decision?<br>• What influenced them to choose one product over the other?<br>• What their peers would choose if given a choice? |
| **Adherence support** | • Whether the adherence support provided helped use the chosen product well.<br>• How did the adherence support differ during crossover and choice periods? |
| **Study acceptability and experience with product use** | • Experience using chosen products.<br>• How products fit in their daily life?<br>• Any challenges with the chosen product?<br>• What made it easier to use the chosen product?<br>• What made it harder to use the chosen product?<br>• Influence from social circles like friends, partners, and family members.<br>• What people in the study said about the products |
| **IDIs included the topics above and, additionally the topics below**. | |
| **Future plans** | How their future plans fit in preserving their health and choosing products to keep them healthy |

participants' product choices were documented on Case Report Forms (CRFs) along with a brief explanation of the reason for the initial choice and subsequent switching (if any).

## Coding and analysis

We iteratively developed a codebook utilizing two analytical frameworks: the Mensch's PrEP/ Microbicide Acceptability Model [28] and the Psychological Empowerment Framework [29] used in developing study guides and codebooks in previous studies [26, 30]. During the codes development process, we iteratively refined codes during weekly team discussions and tested these on several transcripts. Upon codebook finalization, a team of six analysts from the qualitative data management center and one qualitative staff from each research site coded all transcripts using Dedoose software (v9.0.17.) [31]. Intercoder reliability was assessed using the Dedoose training tool to create tests using selected vital codes. A centralized transcript reviewer was assigned weekly to review at least one coded transcript from each coder. Consensus meetings were held weekly until all discrepancies in coding were resolved.

Next, we conducted a thematic analysis [31] from the transcript excerpts coded with the following codes relevant to this paper: product choice (*choice* code) and/or product change (*Switch* code). The *choice* code captured participants' reflections on their decision-making process for the ring versus the pills or neither, including initial thoughts on their projected choice. The *Switch* code included excerpts focused on the participants' decision or desire to switch or stop products. The code reports were exported from Dedoose and included brief summary memos written by analysts, as well as the full excerpts. For the final analytical stage, we applied components of the Protection Motivation Theory (PMT) [32] and a grounded approach to understand these data further and organize new emerging themes [see Fig 3].

Our analysis followed a stepwise approach and a consensus process (through regular meetings and discussions), prioritizing the FGDs as the primary data source on choice, as the timing of the FGDs allowed participants to reflect on their decision-making process for the choice period, including influential factors that contributed to the choice. IDI data were then used to confirm themes from the FGD or identify variations and nuances. We focused on themes that

directly spoke to what participants' decision-making processes entailed at entry into the choice period and the reasons underlying their chosen options. All names used to identify quotes are pseudonyms.

Quantitative data pertaining to participants' characteristics at the start of the choice period and ACASI or CRF data on product choice and switching are presented descriptively.

### Ethics statement/ethical considerations

The study protocol was approved by the Institutional Review Board or Ethics Committee at each site. The study was overseen by the regulatory infrastructure of the United States Division of AIDS (DAIDS) and the MTN. All participants provided written informed consent, and adolescent participants (<18 years old) received written parental/caregiver consent and completed assent forms before participating in the study. All consent forms were approved by the relevant IRB at each site.

### Results

Between May 2019 and May 2021, 247 participants were randomized in REACH, and 119 (48%) participated in the qualitative component. In this analysis, we utilized qualitative data (FGDs and IDIs) collected at or after Month 12, during the *choice period* (Fig 1), for a total of 109 participants. These 109 participants had a median age of 19 years (interquartile range [IQR] 18–20), the majority had completed secondary school (89%), had a primary sexual partner (83%), and were currently sexually active (79%). At baseline [Table 2], 40% of participants in the qualitative sample said they preferred the pills, and 38% preferred the ring for PrEP; 19% had no preference. At Month 12 (start of choice period) [Table 2], more participants

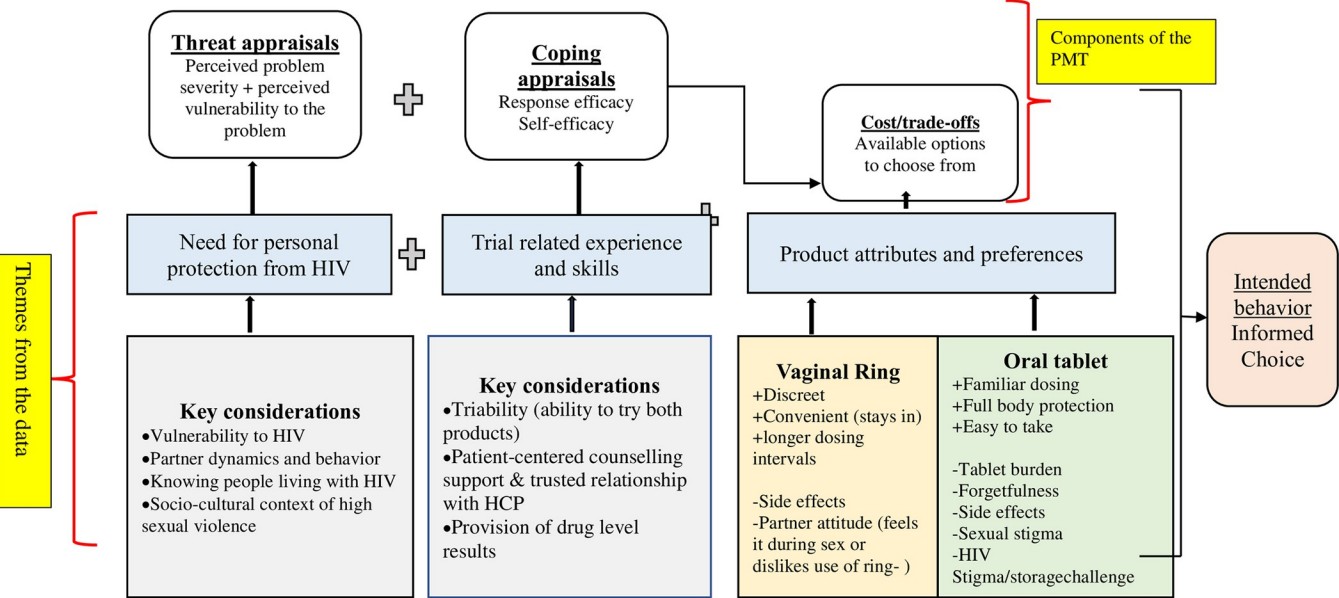

**Fig 3. Key study themes drawing on the protection motivation theory (PMT).** This is an adapted framework of the Protection Motivation Theory (PMT; R. W Rogers 1975). Components of the PMT are represented in the top row of the figure. Briefly, the PMT suggests that environmental and interpersonal sources of information initiate two appraisal processes: threat and coping. The threat appraisal focuses on how an individual's perceived severity of a problem or their sense of vulnerability heightens adaptive or maladaptive behaviour. The coping appraisal focus on response efficacy (knowing that the intervention works) as well as self-efficacy (that an individual is capable of performing the recommended behaviour). In addition to the coping appraisal is the cost/trade-offs of the adaptive behaviour. Key themes and findings from the data are presented in the two lower rows of the figure. For vaginal ring and oral tablets attributes and preferences, (+) indicates positive attributes and (-) negative attributes, respectively.

**Table 2. Qualitative participants' characteristics at start of the choice period (Month 12) and product decisions during the choice period.**

| | South Africa | | Uganda | | Zimbabwe | | Total | |
|---|---|---|---|---|---|---|---|---|
| | N | % | N | % | N | % | N | % |
| Total | 57 | (100) | 30 | (100) | 22 | (100) | 109 | (100) |
| **Demographics & Sexual Behavior** | | | | | | | | |
| Age–*median (interquartile range)* | 19 | (18–20) | 20 | (19–20) | 19 | (18–19) | 19 | (18–20) |
| Currently in school | 30 | (54) | 0 | (0) | 2 | (9) | 32 | (30) |
| Earns income | 9 | (16) | 5 | (17) | 7 | (32) | 21 | (19) |
| Household members(*) | | | | | | | | |
| Mother | 40 | (70) | 12 | (40) | 10 | (46) | 62 | (57) |
| Father | 10 | (18) | 3 | (10) | 6 | (27) | 19 | (17) |
| Siblings | 20 | (35) | 8 | (27) | 2 | (9) | 30 | (28) |
| Children | 4 | (7) | 1 | (3) | 1 | (5) | 6 | (6) |
| Grandparents | 2 | (4) | 1 | (3) | 1 | (5) | 4 | (4) |
| Husband/main partner/boyfriend | 1 | (2) | 4 | (13) | 8 | (36) | 13 | (12) |
| Other | 7 | (12) | 3 | (10) | 1 | (5) | 11 | (10) |
| Lives alone | 2 | (4) | 5 | (17) | 1 | (5) | 8 | (7) |
| Has a primary partner | 48 | (86) | 26 | (87) | 16 | (73) | 90 | (83) |
| New partner in past 3 months | 5 | (10) | 4 | (15) | 1 | (6) | 10 | (11) |
| Any vaginal sex in past 3 months | 49 | (88) | 23 | (77) | 13 | (59) | 85 | (79) |
| Received goods/money for sex- past 6months | 2 | (4) | 11 | (37) | 8 | (36) | 21 | (19) |
| Experienced physical violence from sex partner in past 6 months | 4 | (7) | 2 | (7) | 3 | (14) | 9 | (8) |
| **Product choice at Month-12** | | | | | | | | |
| Ring | 35 | (61) | 23 | (77) | 11 | (50) | 69 | (63) |
| Pills | 19 | (33) | 5 | (17) | 11 | (50) | 35 | (32) |
| Neither | 3 | (5) | 2 | (7) | 0 | (0) | 5 | (5) |
| Top reasons for choice (**): | | | | | | | | |
| *Among those choosing ring (N = 69)* | | | | | | | | |
| Ease of use/fit into lifestyle | 13 | (37) | 16 | (70) | 5 | (45) | 34 | (49) |
| Likes dosing frequency | 19 | (54) | 9 | (39) | 6 | (55) | 34 | (49) |
| Lack of problems or side effects | 8 | (23) | 4 | (17) | 0 | (0) | 12 | (17) |
| *Among those choosing pills (N = 35)* | | | | | | | | |
| Discomfort or concerns with the ring | 4 | (20) | 1 | (20) | 4 | (36) | 9 | (26) |
| Most efficacious /systemic protection | 5 | (26) | 3 | (60) | 0 | (0) | 8 | (23) |
| Lack of problems or side effects | 5 | (26) | 1 | (20) | 0 | (0) | 8 | (22) |
| Ease of use/fit into lifestyle | 4 | (21) | 1 | (20) | 2 | (18) | 7 | (20) |
| **Product switch during choice period:** | 9 | (16) | 5 | (17) | 5 | (23) | 19 | (17) |
| Ring to pills | 4 | (7) | 0 | (0) | 1 | (9) | 5 | (5) |
| Pills to ring | 1 | (2) | 2 | (7) | 4 | (18) | 7 | (6) |
| Ring to neither | 3 | (5) | 3 | (10) | 0 | (0) | 6 | (6) |
| Neither to ring | 1 | (2) | 0 | (0) | 0 | (0) | 1 | (1) |
| Top reasons for switching (N = 19)(*): | | | | | | | | |
| Experienced (new) side effects | 4 | (44) | 2 | (40) | 2 | (40) | 8 | (42) |
| Had use related issues | 3 | (33) | 1 | (20) | 3 | (60) | 7 | (37) |
| Wanted to take a break | 2 | (22) | 2 | (40) | 0 | (0) | 4 | (21) |

(*) While 57% of these participants lived with their mothers, and 17% with fathers, only 14% lived with both parents; 15% lived with >1 adult family member: mother/father/grandparents

(**) open ended responses from CRF coded into categories (and stratified by PrEP product)- top 3–4 ranking categories presented.

chose the ring (63%) over the pills (32%). Based on open response categories in case report forms (CRFs), the top three reasons for choosing the ring were: ease of use, convenient dosing frequency, and lack of perceived side effects. The three top categories for choosing the pills were: systemic protection and/or higher efficacy, ease of use, and lack of perceived side effects. During the choice period, 19 participants (17%) switched or stopped using PrEP products. The main three reasons for switching or stopping (per CRF responses) were: experienced (new) side effects, user-related challenges and wanting to take a break.

Cross-cutting themes were identified from FGDs and IDIs for a) choosing a product versus not and b) choosing between the ring or pills. The following sections present overarching themes and specific sub-themes that emerged from the data, along with exemplar quotations [see Fig 3]. The first theme, *protection from HIV as a priority*, reflected a combination of individual and social-contextual factors that were salient in influencing participants' choice to use PrEP. The second theme reflected *trial design factors*, which included product triability (ability to try the products before choosing) during the crossover periods, and other study components (e.g., client-centered counseling, drug level feedback). The third theme reflected *product attributes* and their impact on participants' experience and selection process. Altogether, these themes, in combination, informed women's decision-making processes and enacted preference during the choice period.

## 1. Protection from HIV as a priority

Almost all participants chose one of two offered biomedical products at entry into the choice period. This decision was largely influenced by their perception that they had a high likelihood of exposure to HIV and wanted to prevent acquisition. Some also reported first-hand experience with individuals living with HIV or who had died from AIDS, as Teddy emphasized: '*I still wanted protection because there is nobody who wants to be infected with HIV; there is no one. What forced me to use these methods is that I have seen people infected with HIV in my family; some of them I was taking care of, and I buried some of them. . .*' (pill choice, FGD01-Uganda).

Furthermore, participants shared other vulnerability considerations that played a part in their product choice decision (see Table 3-as additional supporting data), including i) inconsistent condom use coupled with participants' acknowledgment of their inability to negotiate condom use with their partners and having little control over when condoms were used or how well they were used, ii) partner dynamics, which encompassed mistrust of partner behaviour relating specifically to their sexual conduct outside of the relationship, and iii) living in contexts of unbalanced gender power dynamics and sexual violence. Indeed, forced or coercive sex, which is typically unplanned, manifested through multiple reports of rape (in Cape Town, South Africa) as well as unsafe transactional sex encounters (in Uganda), emerged as strong motivators for participants' product choice decisions.

## 2. Choosing the right biomedical option through experiential learning and product triability

REACH participants often described how trying both products during the crossover period of the study was central in shaping their choice at Month 12. Five interrelated sub-themes that describe their views were identified: **i)** weighing what fits them best; **ii)** extinguishing initial fears around products; **iii)** debunking local myths about PrEP as they gained familiarity with the products; **iv)** re-appraising their default options when choosing and **v)** utilizing knowledge and skills obtained from trial activities, such as support strategies (i.e., counselling, adherence clubs, and drug level feedback) when making their choice. These are described in detail below.

## i)Weighing what fit them best

Across the three countries, and regardless of their choice of product, participants evaluated their personal experience with each product and which fit best for them in terms of convenience, lifestyle, and side effects, as exemplified by Jennifer: *'When I was going for my choice [product], I first had to think and see what had worked for me best. The ring is not bad because for the pills, there comes a time and[when] you forget to swallow them, or when I swallow them, I get some problems and get some[a] headache. And [when] I started using the ring as the other person has said, the ring has that problem of bad smell and fluids [discharge] come from your private parts, and by the time you remove your knicker, it is dirty even if you have not been putting it on for a long time'* (ring choice, FGD02-Uganda).

Other women described how the trial gave them the confidence to choose with ease, as they had tried both products: *'One, it gave me confidence and having them choose for me [crossover periods] gave me more time to experience those things[products] so that when it came to choosing, I knew what to choose with confidence. That is how it helped me make a decision. . .obviously, you can't choose something that you have never used, isn't it?'. . .*(Lucy, ring choice, FGD04-Zimbabwe). Conversely, 15 women who chose neither product at Month 12 reported major barriers with each product that they could not overcome: *'The pills do not cause a bad smell [like the ring], but they do cause dizziness and nausea [. . .]Since I had used both of them, I had an experience with both, but I didn't fulfil what I was supposed to do [use them correctly]. Then I thought I may decide to take the vaginal ring, and I don't use it well, then*

**Table 3. Factors influencing choosing a product or neither (additional data).**

| PMT Aspect | Theme | Sub-theme | Example Quotes |
|---|---|---|---|
| Threat Appraisals | Need for Personal Protection from HIV | Vulnerability to HIV | *Prevention is always important, that is why I ensured that I used a product so that I am protected. My health is important.* (Vimbai, ring choice, FGD02-Zimbabwe) |
| | | Partner dynamics and behavior | *The fact that we don't live together, I live here, and he stays there it can happen that I am worried of him somehow. you know it is not easy to trust someone.* (Nabada, pill choice, CIDI-Uganda) |
| | | Interaction with People Living with HIV | *I feel like there is a lot of risk. First time when I came to the study, I wasn't with the partner that I'm with now, my ex-partner was positive, and I discovered that he was positive when I was deep into the relationship and we did not use a condom and a girl found out that she was positive because of this guy.* (Kit, ring choice, FGD01-SA-Johannesburg) |
| | | Socio-cultural context of high sexual violence | *Because us women where we make our movements a lot happens, and you can't be safe but with this study, if you still have a chance you can use. So, in this case if you happen to get any problem you have got protection, I wanted to have protection that is why I chose this ring.* (Melissa, ring choice, FGD03-Uganda)<br>*Like to take in everything that my body has been getting, the ring and the pill. So I told myself that I will take PrEP, since I had already been introduced to what PrEP does. Not because I am at a risk of having HIV, no. But to protect myself because as girls, you can be just raped. So I will take them from the clinic.* (Mimi, chose neither, FGD04-SA-Cape Town) |
| Coping Appraisals | Trial related experience and skills | Triability (ability to try both products) | *I felt good because you are given options. A chance to see which one will work better for you, that will be easy for you to use and be comfortable with; so that you can be able to choose for yourself.* (Pretty, chose neither, FGD01-SA-Cape Town)<br>*Yes, when we were at choice stage, it helped [to have tried both] because I noticed that the ring-, the ring caused side effects and the pills [PrEP] did not.* (Pill choice, SIDI3-Zimbabwe)<br>*When I reached that stage of choice, I felt good because I used the ring as well as I had used Truvada [the pill]. Now I could make a decision, so I had to make my choice.* (Namusoke, ring choice, FGD03-Uganda). |
| | | Patient-centered counseling support & trusted relationship with healthcare provider | *For me I had raised that question to the counselors that I fear using pills, so they had to call mum at home and they told her that she has to always make me take that thing [Pill] when she is seeing, and then when I took the stuff, it [Pill] was not all that big, I was thinking it [Pill] is big. So, I had to ask them, "can I use sweet bananas when I am swallowing?" I was told that there was no problem, then that fear went away from me, the fear of the ring also went, the fear of Truvada also after seeing Truvada went I was able to use Truvada, at the same time used a ring well.* (Eva, ring choice, FGD03-Uganda) |
| | | Provision of drug level results | *What made to go to a ring is the drug levels. I was on green all the time and I thought let me choose since I was always yellow with the pill. So I thought what will help me, let me choose the ring again that I am using well.* (Vuyo, ring choice, SIDI3-South Africa) |

*(Continued)*

**Table 3.** (Continued)

| PMT Aspect | Theme | Sub-theme | Example Quotes |
|---|---|---|---|
| Cost/Trade-offs | Ring: Product Attributes and Preferences | Discreet | *I chose the ring because I always have it on me, and I won't forget it like I did with the pills. Also, I will not feel ashamed having to travel with the pill bottle; but I will have always the ring, and no one will know that I have a ring.* (Lutendo, ring choice, CIDI-Zimbabwe)<br>*Because I don't have to move with it, it is inserted in the female genitals, no one knows about it even when you walk no one knows what you are using because it is not accessible. But you may have a bag and when a friend checks it out she may see the pills.* (Nasolo, ring choice, SIDI3-Uganda) |
| | | Convenient | *Once you have it you can even forget that you have it until you are removing it, but for the pills you have to take them every day.* (Natasha, ring choice, _FGD02-Uganda)<br>*As I explained before that the ring is always on you, you don't take it out, as for the pills you may forget them. I found the ring easy to use because it is always on me, I don't remove it or anything.* (Tanya, ring choice, FGD01-Zimbabwe) |
| | | Side effects | *The ring makes that bad smell and you have all the time to be clean, so if you don't have enough hygiene then it can lead you to smell.* (Eva, ring choice, FGD03-Uganda)<br>*The ring is just too big and uncomfortable, and smelly.* (Sindi, Pill choice, CIDI-SA-Johannesburg) |
| | | Side effects (con't) | *As for me, I did not choose the ring because it caused vaginal fluids. But as for the pills there is nothing bad I experienced from the time I started taking them. This is because with the ring I could bath even three times in a day otherwise if you did not, people will shun you saying you are stinking. So I find that pills are better because I can take them without any issue, that is why I chose the pills and left the ring.* (Martha, Pill choice, FGD02-Zimbabwe) |
| | | Partner Attitudes | *As for me what made me to leave the ring is that my partner would feel it and it was coming out. So I said to myself, "Ah it is better I leave the ring and take the pills". I found the pills better for me.* (Farai, Pill choice, FGD02-Zimbabwe)<br>*He asked me "What is this?" and I told him that it was a ring but I didn't tell him whether it was preventing HIV…When you tell some they think you are a prostitute "Do you have someone else?" He might think that I am a prostitute. He knows that you only have him [But when you tell him about the vaginal ring] he will think you have other sexual partners.* (Namaganda, ring choice, CIDI-Uganda) |
| | **Oral Pill:** Product attributes and preferences | Burden | *The ring does not give me problems. I don't struggle like, "yoh! I'm already asleep, let me wake up and drink the pill". And the water is cold at this time for me to be drinking the pill.* (Vuyo, SIDI3, ring choice, SA-Cape Town)<br>*I chose the ring. The ring is not the same as pills. With pills, you are always worried of [about] swallowing them every day, and [it] was giving me a lot of headaches and I was always in the clinic and could also go to [the] hospital, and I felt that I was fed up. That is why I decided to use the ring because it does not have a lot of problems except for you feeling that you are not clean. Even if you are seated among the people, you can think that you are smelling. That happens to me as a person, and I chose to go with it.* (Jennifer, ring choice, FGD02-Uganda) |
| | | Forget-fulness | *So I see that pills have various challenges. That's why I chose to use the ring, because even when you are traveling you may forget to put that key holder [referring to the pill case] in your bag. That's challenging using them [PrEP pills]. It's possible to take pills every day but it's a bit difficult compared to using a ring.* (Faith, ring choice, SIDI3-Zimbabwe) |
| | | Forget-fulness (con't) | *You can forget to swallow the pills and go for work but the ring is already there and so you are not on pressure.* (Jennifer, ring choice, FGD02-Uganda) |
| | | Familiarity, easy to use | *I do not feel comfortable because it will be inside of me That is why I do not like it. The pills, I like them because I am used to taking them.* (Chloe, Pill choice, FGD04-Zimbabwe)<br>*Since I am someone who had never seen the vaginal ring I would choose pills.* (Tanya, Pill choice, SIDI3- Zimbabwe) |
| | | Side effects | *I thought the pills were not good for me from the start because when I drink them, I would vomit and, not feel well, and be sleepy, like my body would be tired. I thought no, I am not going to be able to stand for that because I like going out and I like to be fresh. So I prefer the ring because ever since I have used the ring it has never troubled me.* (Emihle, ring choice, FGD03-SA-Cape Town)<br>*Because the pills caused me nausea and I feel like I want to vomit, so I am forced to eat clay or soil [to treat the side effects] yet, it is not good for my health, but I have to protect myself, and yet the ring makes me feel not clean, all the time I feel I am smelling, the fluid [discharge] that comes out smells very bad even when you put off the knicker you see it is very dirty and that fluid that comes out is very dirty. As if I had no choice but [to] choose pills because I wanted protection.* (Kayson, Pill choice, FGD02-Uganda) |
| | | Sexual stigma | *Some of my peers are very judgmental; they think if you are using PrEP or the ring, you are gallivanting, you are sleeping with different men and stuff, they don't think is for protection, they have different views.* (Nobuhle, ring choice, FGD03-SA-Johannesburg) |
| | | HIV stigma | *Some of the people in our household, they would asked me as I was taking the pills and say, "Could it be that you are now taking pills for AIDS?* (Martha, ring choice, FGD02-Zimbabwe) |
| | | Storage challenges | *There is a challenge in forgetting when travelling, and the pills [PrEP] are something that everyone can just notice if you store them in the house.* (Faith, ring choice, SIDI3-Zimbabwe) |

*also decided to take the pills but end up not using them well, that's why I decided not to use any method'* (Nansubuga, chose neither, NIDI-Uganda)

### ii) Extinguishing initial fears around the products

A theme strongly conveyed in participants' narratives was how the study crossover periods obviated their initial fears about the pills and the ring, thus enabling them to choose products with ease. This was specifically in reference to the ring being an unfamiliar product. Participants described how they were afraid of its size, its insertion into the vagina, how it would feel in situ, and whether it would fit, as Zamajoba explained: '*Yoh! I was scared of the ring. When I was sitting there, I was like, why is this thing like this because I had never used a female condom? So, I was really scared of it and figured that I would rather take the pills because I would never insert this thing underneath [in the vagina]. But then when I inserted it and saw that I did not have a problem, I liked it*' (ring choice, SIDI3-SA-Cape Town). For some, their fears about the ring were based on other participants' descriptions and experience of the ring: '*As for the ring, I heard others saying that it is painful and that it makes one to release a lot of [vaginal] discharges'* (Rose, ring choice, FGD01-Zimbabwe).

### iii) Debunking rumors or myths surrounding prevention products

Aside from the benefits of trial participation that enabled women to confront their own fears, the crossover periods further enabled participants to challenge myths and rumors that would have otherwise thwarted their abilities to choose either product. Participants described how the experiential knowledge and information gained helped them to challenge deep-seated and long-standing myths or rumors about HIV prevention prevalent in their communities and widespread amongst their peers, including associations of the products with having multiple and/or concurrent partners and HIV status misattribution: '*Some of my peers are very judgmental, they think if you are using PrEP or the ring you are gallivanting, you are sleeping with different men and stuff, they don't think it is for protection, they have different views'* (Amahle, pill choice, FDG03-SA_Johannesburg). Another participant shared a different concern that made her resist inserting objects into her vagina as she was told by her friend that '*inserting products in her vagina would cause her cervical cancer'* (Latifa, ring choice, FGD03-Uganda). During the crossover periods, however, participants quickly learned as they used both products and through discussions with peers in the trial and regular interaction with clinical staff that these were simply falsehoods spread in their communities that didn't align with their lived realities.

### iv) Re-appraising their default option

Participants across all sites expressed similar preconceived notions about each product and thought they knew what they would choose from the outset. Most participants described how they would have defaulted to the more familiar pills had they not been given the opportunity to try the ring: '*I don't think anything would have changed; maybe the first time period, I would have chosen for myself, I would choose the pills because like she said, they looked more familiar to us than the ring'* (Star, pill choice, FGD03-SA_Johannesburg).

Similarly, upon trying both products, several participants reconsidered their initial preferred option, not realizing the burden of daily dosage and that it would have been a mistake to opt for pills. For example, prior to using either product, Vimbai highlighted that: '*I would have chosen the pills, but once I used the pills, I had some challenges' (ring choice, FGD02-Zimbabwe)*

Furthermore, participants indicated that without actual experience, if these products are rolled out without the opportunity to try each like in REACH, clients would choose the pills as their default option: '*Our peers would want the pills because they already know about the pills,*

*they [pills]are there in government facilities, and they are taught about them by the health workers, but they are not aware of the ring*' (Rita, ring choice, FGD03-Uganda).

Nobuhle, like Amahle quoted above, mentioned that without trying the products first, peers outside of a trial setting might decline all PrEP products, fearing sexual stigma: '*Because we live in a society that is too judgmental, people in our age group think that when you [are] taking a product like this you are exposing yourself to boys and you like to change boys [having multiple partners], I think they would choose nothing unless they [are] attending in a study [like REACH] and have more information*' (Nobuhle, pill choice, FGD03-SA_Johannesburg).

## v. Trial support strategies

More subtle but significant factors influencing participant choice were the in-person counseling and drug-level feedback received in the REACH adherence support context, as described below. These educational and skills-building sessions taught participants what to expect from each product which helped them make autonomous decisions.

Participants highlighted how the counselor's participant-centered and problem-solving approach helped them navigate through their experiences with side effects during the cross-over period, a skill they were able to employ in the choice period with their chosen product: '*Every time when I go for counseling with her. . . she asks me to come up with solutions all the time that I personally think will help me in my situation. She always looks for solutions that are best for me, so she does not choose solutions for me; she just helps me in suggesting [by suggesting]. . .with the choice period I loved it. I knew exactly [what to do], [but] by the time the choice period came, I had already decided*'. . . (Kit, ring choice, FGD01-SA-Johannesburg). The availability of counselors to help participants cope with side effects increased their self-efficacy both to facilitate choice and persist with the products through difficulties.

Also, for most participants, drug level feedback (DLF) served its intended purpose as an adherence metric (how well they used a product). Furthermore, it helped as a tool to confirm that their choice was **correct** by showing that they were achieving high levels of protection with a product they perceived that they were able to use well. For Vimbai, DLF signaled how adequately protected they were while using a product:

> '*Yes, I say my drug level results influenced the product I chose because I forgot the pills, but with the ring, once I insert it, I would be protected for the whole month. Compared to the pills, if you forget, you will be protected less. That is why I chose the ring because of the drug-level results I received. When I noted that I was in yellow, I observed that it is not helpful to choose the pills because I am not fully protected, but I had no challenges with the ring*' (Vimbai, ring choice, FGD02-Zimbabwe).

Through DLF and the adherence support they received in REACH, participants also realized that it was important to choose a product that fit their lifestyles to maximize prevention and effective adherence:

> '*And then, like, when I have forgotten it [the pills], it will not protect me. It won't be 100% because I usually forget the pills. So, I thought if I [have] inserted the ring, I am 100% sure that it will protect me more than the pills that I usually forgot. It [the ring] is in my system. It would not be as strong [but it's better than] me forgetting it[to take the pill]*' (Zamajoba, ring choice, SIDI3-SA_Cape Town).

Namuli, who chose neither product, highlighted that '*: Basing [based]on the colors that I used to score, I decided at once and said "let me leave both of them"*' (chose neither, NIDI-Uganda)

Other participants interpreted DLF as a measure of what was **convenient** for them to use. In most cases, participants reported having green results while on the ring, mainly because it was always inside. They reflected on the results they obtained with each product and determined that dosage regimen determined their results. For the pill, yellow or red results were linked to their struggles with daily dosing, which enabled them to make concrete decisions of what product to choose: '*In my first six months, I was using the pills. With the pills, it was not the same. Sometimes they would be green and sometimes yellow. Only with the pills. With the second six months when I was using the ring, I always received green because I did not remove the ring, it was always inside me. It was always green, green; green. Like also having to choose, I chose the ring now*'. (Connie, ring choice, FGD04-SA_Cape Town). Esther echoed similar sentiments: '*I liked it when I had options to choose [referring to study products], the pills and my Wi-Fi did not get along. It wasn't ideal for me but as for the ring its going along with the Wi-Fi [DLF], like the category that I am supposed to be in*' (ring choice, FGD03-Zimbabwe).

Yet other participants highlighted that DLF did not influence their choice: instead, they based their choice on the perceived advantages of the products. For example, some participants liked that the pills protected the whole body, and despite receiving yellow DLF results, they went for systemic protection. [see Table 3].

## 3. Product attributes and preferences

### The ring

As detailed in Table 3, several sub-themes emerged from participant narratives surrounding product attributes and characteristics influencing choice. These themes were organized as follows: (i) The ring delivery mechanism, which involves insertion into the vagina where it releases the dapivrine drug slowly over a month. Participants who chose the ring described this as convenient, simple, and easy to use and bringing about '*a peace of mind*' (Nombole, ring choice, SIDI3-Uganda) as participants never have to worry about it once in place. (ii) Ring discreetness: participants reported the invisibility of the ring once inserted as a critical advantage, enabling them to use it covertly without anyone knowing. Ring invisibility and discreet use appeared to occur at various levels: firstly, participants reported inserting the ring and completely forgetting about it: '*So with the ring, you insert the ring once; here at the site and then you forget about it*'. . .(Connie, ring choice, FGD04-SA-Cape Town). Second, some participants reported choosing the ring to avoid disclosing it to their partners, who they feared would judge them or even prevent them from using it. Finally, the ring was discreet to family and social networks. Kayson explained that: '*Because according to the people I am staying with, in case you have the pills, they might think that you have HIV. But for the ring, nobody is going to see it or know that you have it, and you can have it as long as you need it*' (ring choice, FGD02-Uganda).

Indeed, several participants considered their living environments when choosing a product, as they lived in settings with little privacy: '. . .*the pills were a challenge because at our home there were a lot of people. So the people would go about saying bad things about the pills, so that is why I chose the ring*' (Kaseke, ring choice, FGD02-Zimbabwe). Participants who chose the ring shared that compared to the pills; one could use the ring stealthily until one was ready to disclose. The fear of being judged or labeled as living with HIV while using the pills was thus avoided by choosing the ring. At the community level, participants described shame and stigma often associated with taking the pills. Conversely, the ring was discreet enough for the social environment to avoid detection: '*I will feel ashamed having to travel with the pill bottle; but I will always have the ring, and no one will know that I have a ring*' (Lutendo, ring choice, CIDI-Zimbabwe).

Barriers to choosing the ring often included side effects (pain in the abdominal area during sex, excessive lubrication, and smelly discharge) and type of protection (vaginal protection only; see Table 3). But here, too, there were some tradeoffs so that certain side effects were weighed and accepted for the benefit of the ring's convenience and protection.

### The pills

Participants, especially those in Uganda and Johannesburg, SA, reported choosing the pills because it provided full body protection: '*My choice was the pills, the vaginal ring was very okay, I didn't get any problem with it, so the reason why I chose the pills is that they protect the whole body . . . but the vaginal ring protects the vagina only*' (Teddy, pill choice, FGD01-Uganda).

'*. . .For me, [in] at the last six months, I chose the pills even though I like the ring, I did not choose the ring because it only protects you when you have vaginal sex and the pills protect your whole body.*' (Blue, pill choice, FGD02-SA-Johannesburg). Several participants in Cape Town and Zimbabwe described opting for the pills if they did not experience any side effects or simply because they liked and found it familiar to use: *For me, I found the pills better for me. My partner used to feel the ring*'. . .(Farai, pill choice, FGD02-Zimbabwe). Imani from Cape Town shared similar sentiments: '*I choose the pills because the way I see it, I do not have a problem with the pills. It is easy to swallow them, and they do nothing [no side effects] on me*' (pill choice, FGD02-SA-Cape Town). Some, like Blue, identified practical strategies to remember taking pills on a daily basis: '*I get reminded by my mother because she takes the ARV's so every time, every day, I have to make sure that when I give my mom pills, I also take my pills*' (pill choice, FGD02-SA-Johannesburg). In contrast, barriers to choosing the pills were often linked to user burden, disclosure challenges, poor persistence, side effects, and its association with sexual and HIV stigma (see Table 3)

### Discussion

Findings presented in this paper are drawn from 109 participants in SSA who participated in the first-of-its-kind, 18-month MTN-034/REACH trial that allowed participants to experience both the ring and the pill for six months and then offered them an opportunity to choose their preferred option during the last six months of the study. Most participants chose one of the two products, with the majority opting for the ring. Most participants reported the desire for protection, the opportunity to try both products and the ability to weigh between rings and pills' attributes as important factors influencing their decision-making and choice.

Unlike many AGYW studies highlighting poor persistence occurring three months after initiating oral PrEP [33, 34], participants in REACH were highly motivated, completed 12 months of product use during the crossover periods, and most continued for an additional six months with a product of their choice [21]. First and foremost, participants wanted effective HIV prevention. The motivation to choose a PrEP product (vs. none) is consistent with HIV literature that generally attributes one's perception of risk to the adoption of HIV prevention interventions [35, 36]. This also fits well within the protection motivation theory (PMT) threat appraisal, which suggests that various contextual and interpersonal sources of information initiate processes that enable individuals to devise ways to engage in protective behavior, especially if those processes exert a degree of fear [like fear of getting HIV] on individuals. Here, participants highlighted the fear of acquiring HIV, living in settings of high HIV prevalence, and knowing people affected by HIV as important decision for staying on PrEP during the

choice period. They further reported inconsistent condom use[37, 38], non-monogamous partners [39], and gender power imbalances that encourage sexual violence as threats, shaping their decision to opt for PrEP [40, 41]. Although threat appraisal is necessary for the process of behavior change, it is not sufficient to change behavior. The PMT considers coping appraisal, which is a combination of self-efficacy and response efficacy [knowing that an intervention works], as a necessary mechanism that augments the threat appraisal to lead to behavior change [32]. Thus, the majority who opted to persist with PrEP in the choice period were influenced by the perception that they had a high likelihood of exposure to HIV [35, 36] and, additionally, empowered by the knowledge and skills gained during the crossover periods in REACH. Indeed, a key facilitator participants reported was the ability to try the products before choosing. 'Product triability' 'Product triability' in this paper means the degree to which an innovation may be experimented with in a controlled environment to enable choice decisions helped participants in several ways, including demystifying unfamiliar products (mostly ring) and minimizing the perceived risk of choosing the "default" product because it was familiar (mostly the pills). By the time they chose, participants knew what best fit their lifestyle. Triability also allowed participants to weigh pros and cons based on personal evidence rather than relying on rumours or vicarious experiences [42]. Triability further enabled participants to dispel fears and preconceived notions about the products, mitigating the impact of ongoing rumours on adherence. Finally, triability built confidence in participant decision-making processes based on their ability to use the products well (or not). What is described above is consistent with broader HIV literature and other health-related fields (e.g., smoking, alcohol, and drug use) where adopting a protective behavior is driven strongly by perceived self-efficacy and results from a cost-benefit balancing act, both key components of the PMT, which participants in this study obtained through personal experiences, study processes and counseling-related activities [43]

Although triability, as in the REACH trial, is hard to implement in programmatic settings, our findings indicate it is essential that PrEP *options* are provided to end-users and that they should be supported to choose and switch methods, as choice is dynamic. Furthermore, practical approaches to triability can also improve informed decision-making, as previously reported. These include trying a product at the clinic under direct supervision as in REACH and other studies of oral PrEP or vaginal devices, using a pelvic model or other visual props to understand product administration and placement, and/or videos with product demonstrations and testimonials, which all can offer brief but important exposures to products before choosing [42, 44–46]. We posit that incorporating some triability options programmatically is important, especially for user-controlled and fully reversible PrEP methods like pills and rings, although how these can practically be best 'tried' in a clinical setting may need to be further assessed through demonstration projects. Finally, if individuals are not able to try products themselves, triability, as seen in the lens of Rogers' diffusion of innovation theory [32], where those who have tried the product share their experience and function as product ambassadors for naïve users, can be the "next best" approach that may enable the broader community to adopt these new PrEP options [47].

In choosing one PrEP modality over another, the dominant difference was the dosage frequency with a preference for less frequent dosing: "Set it and forget it" [48, 49], which decreases users' daily dozing burden. Of note, while forgetting a Pill was a negative attribute of oral PrEP, inserting the ring and forgetting about it was a positive attribute for the ring. This may be one critical differentiator to highlight when counseling individuals about these options in programmatic settings. Convenience and comfort were also key to preference, while lack of side effects was a key motivator for choice. The tolerability of side effects, strategies to mitigate/manage them, tradeoffs between these, and the beneficial attributes of the products were

weighed by participants during their decision-making process. Similar to previous literature, in REACH, side effects were a primary driver of dissatisfaction and discontinuation of methods [50, 51]. In REACH, about 10% of the women who used the ring reported increased vaginal discharge (then referred to as vaginal fluid) as a side effect, and for some, this changed their preference for the ring. Thus, as previously reported, product choice in REACH was a "push-pull" balance between preference for key attributes of one product and aversion to the attributes of the other [27]. Several other external factors were also weighed by participants when choosing, including privacy, support by key influencers, work constraints, resources, mobility, sexual, and HIV stigma. Thus, discretion was often key, given the high social stigma about HIV in these settings [52]. Furthermore, discretion was mostly discussed relative to the specific circumstances and sexual situations of participants (e.g., living with a partner or not for Pill bottle storage, partner feeling the ring or not during sex).

Notably, nineteen participants switched their product during the choice period despite having had six months of use with each during the crossover periods. Some switches resulted from (new) side effects with their chosen product or because their personal circumstances changed, creating new barriers to using a particular dosage form. In the TRIO study, switching was common as well [20]. This highlights that product discontinuation, pausing or switching are normal behaviors, as previously noted for oral PrEP and contraceptives, and will persist even if product triability is possible [53, 54]. Thus, recognizing that choice is dynamic, providers should embrace method change as an inherent part of HIV prevention. Hence, developing strategies that support users in product testing, trying, and changing to identify the best current fit is key as more options become available for PrEP.

The provision of adherence support strategies (including DLF) and counseling during REACH were not cited as pivotal factors influencing choice. Yet, participants indicated that these enabled them to make more autonomous and empowered choices. Indeed, shared decision-making in clinical settings supports patient-centered informed choice and is increasingly recommended for PrEP [55]. In REACH, the client-centered approach enabled creative troubleshooting, personalized solutions, and autonomous decision-making. For choice, unlike adherence, this may be especially important early on in the user journey, but once mastery of product use is achieved, it might be less critical. Beyond professional support, building peer support might be equally important for social acceptance and sharing similar challenges and tried-out solutions. [56]

Our study notes the following limitation: AGYW taking part in REACH had good prior knowledge about PrEP (through study-related recruitment activities) and were required to use long-acting contraception for at least a month before starting their first crossover period, resulting in a substantial selection bias at enrolment for compliant participants. As frequently noted, with qualitative research, purposing sampling may have led to selection bias. Furthermore, qualitative methodology is interpretive–However, we minimized recall bias by ensuring that the data were collected close to the time of choosing while the process for decision-making was still fresh. In addition, all interviewers were external to the trial implementation activities to minimize social-desirability bias. Also, because of the COVID-19 pandemic, we minimized the number of FGD participants to allow for social distancing. We cannot assess how this affected the group dynamic. We also collected in-depth interviews that supplemented the FGD data presented.

In Sum, the REACH study demonstrated that young people, including AGYW, can be supported to use oral and vaginal PrEP, are able and eager to make their own decisions about best-fit products, care about being protected with minimum social inconvenience and life disruptions, and value choice of options.

## Conclusion

In REACH, participants successfully tried and used two PrEP options and selected the right fit for them when asked to choose. Key themes around choice identified in this analysis can inform programmatic guidelines for each product when offered side-by-side in clinic settings. It will require developing client-centered messaging, clear and factual information about each product–including side effects-, an opportunity to try and switch products based on initial experience, and support via joint decision-making with health providers throughout the choice process, along with peer support to facilitate social adoption of PrEP.

## Author Contributions

**Conceptualization:** Millicent Atujuna.

**Data curation:** Alinda Young, Erica N. Browne, Nomvuyo T. Mangxilana, Mary Kate Shapley-Quinn.

**Formal analysis:** Millicent Atujuna, Erica N. Browne.

**Methodology:** Millicent Atujuna.

**Supervision:** Ariane van der Straten.

**Writing – original draft:** Millicent Atujuna, Kristin Williams.

**Writing – review & editing:** Sarah T. Roberts, Alinda Young, Siyanda Tenza, Mary Kate Shapley-Quinn, Thelma Tauya, Kenneth Ngure.

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
