## [Decision Letter · Decision Letter 0]

2 Nov 2023

PONE-D-23-24059We Choose : Adolescent girls and young women’s choice for an HIV prevention product in a cross-over randomized clinical trial conducted in South Africa, Uganda, and ZimbabwePLOS ONE

Dear Dr. Atujuna,

Thank you for submitting your manuscript to PLOS ONE. After careful consideration, we feel that it has merit but does not fully meet PLOS ONE’s publication criteria as it currently stands. Therefore, we invite you to submit a revised version of the manuscript that addresses the points raised during the review process.

The manuscript delves into the critically important area of HIV prevention among young women in the South Africa region, a topic that holds significant relevance in contemporary public health debate. The authors have presented a commendable effort in shedding light on this subject. However, to ensure that the manuscript meets the rigorous standards of our journal, I suggest undertaking a thorough review and making necessary revisions. This will not only enhance the clarity and depth of the research but also align it more closely with the journal's established criteria.

We look forward to receiving your revised manuscript.

Kind regards,

Ivan Alejandro Pulido Tarquino, MSc

Academic Editor

PLOS ONE

“The work presented has not been published and is not under consideration in any other peer-reviewed media. The listed authors have all contributed significantly to the design, analysis, and written work, and all authors have given final approval for the version to be published. To the best of our knowledge, no conflict of interest, financial or other, exists.”

Additional Editor Comments:

The paper offers a comprehensive overview of the advancements made in HIV prevention, focusing on a subject of great interest from the perspective of the beneficiaries. Nonetheless, to increase the manuscript's quality, I believe the authors should heed the suggestions provided by the reviewers. Moreover, I'd like to emphasise the following points:

1. It's crucial for the authors to provide a scientific rationale for the sampling techniques employed and clarify the basis for determining the final sample size. Which specific criteria or references from prior studies influenced this decision?

2. For clarity, I suggest the results section include a well-organized table detailing the characteristics of the FGDs. This table should outline the number and origin of participants for each FGD, address potential biases due to participant demographics (e.g., all participants of the same age? younger (16) vs older (21)), and specify the FDG locations (e.g., healthcare facility name, county, or neighbourhood). Upon examining Table 2, it's challenging to differentiate between FGD and IDI participants.

3. To prevent confusion, the term "private place" inside the healthcare facility needs to be defined precisely.

4. I've noticed the authors refer to "tablet" while interviewees say "pills." The authors should either choose one for uniformity, which is, in my opinion, a better option, or maintain this distinction throughout for consistency.

5. Based on the findings, the discussion and/or conclusion sections should include more specific recommendations about the implications and continued implementation of HIV programmes in the study sites. Adding more detail to this would improve the paper's clarity and comprehensiveness.

Reviewers' comments:

Reviewer's Responses to Questions

**Comments to the Author**

1. Is the manuscript technically sound, and do the data support the conclusions?

Reviewer #1: Yes

Reviewer #2: Yes

2. Has the statistical analysis been performed appropriately and rigorously? 

Reviewer #1: Yes

Reviewer #2: N/A

3. Have the authors made all data underlying the findings in their manuscript fully available?

Reviewer #1: Yes

Reviewer #2: Yes

4. Is the manuscript presented in an intelligible fashion and written in standard English?

Reviewer #1: Yes

Reviewer #2: Yes

5. Review Comments to the Author

Reviewer #1: This is a very well written paper on a highly relevant topic.

Recommendations for revisions:

Major:

Intro:

Please spell out more clearly how the study is novel compared to previous studies. The argument “However, research examining product preference and choice using active products is still lacking” seems to contradict with the previous sentences (“found that discreetness and longer duration of protection to minimize user-burden were favored, along with products that did not interfere with intimate relationships, and provided HIV prevention for unanticipated situations (18). Furthermore, previous experience with a long-acting contraception (e.g., implants) influenced choice of product”) showing factors that inform decision-making.

Methods:

What determined the number of FGD and IDI? Was any interim analysis conducted?

How did you validate? Was any triangulation (eg of data obtained from different stakeholders) done?

Table 2: check formatting of lines

Results:

A table describing characteristics of qualitative study participants next to those included in the trial but not in the qualitative study would be relevant

Discussion

Emphasize novelty of findings shown, if any. The first paragraph is a good place to do this.

441-457: few references – reads like the results section. 457-461: requires refs

Discuss drug level feedback: how could this (or another approach aiming at the same type of info “yes, you ‘re doing well”) work in a programme?

Limitations: reflexivity is recommended: how did the experiences and views of the researchers and moderators affect data collection and analysis? Any other bias? Recall bias? Social desirability in a trial on new PREP methods?

Minor:

Abstract: “use,”: comma should be point

Intro: “understand how the use of active products may pose different challenges compared to prior findings in placebo trials” could be “how the comparison of active products” and “placebo-controlled”?

Reviewer #2: Firstly, thank you for the opportunity to review this manuscript. It is very well-written, the research design itself is clear (coming from part of a well structured, well-known larger study) and the findings are helpful and have an operational relevance to many different contexts. One of the most interesting parts of the study design is the ‘crossover’ period and I think that is something readers can really learn from, so would like to see more made of this in the results.

1. Introduction and methods

Can the authors offer a definition of what is meant by 'modern contraceptives'?

This is an interesting study design which clearly a lot of thought and expertise went into - well done! I have made a few comments and suggestion for clarification below.

The authors state that a total of 16 FGDs were conducted, and if I understand the table correctly, a total of 62 participants were involved. This means that each FGD only had between 3 and 4 participants in it. These low numbers pose a methodological challenge as the group dynamic would not be large enough for the benefits of a focus group, which should typically be around 8-10 people if carried out according to qualitative research guidelines. Please can the authors include these low numbers of participants as a limitation and explain what was done to mitigate this, reflect on why there were so few participants, and whether group interviews were considered instead?

The tool used for FGDs is described as an interview guide - should this be ‘focus group guide’, as the questions were amended to reflect the group context, and weren’t identical to the interview questions. The distinction between the two guides is made in the table, but not in the narrative.

The process for recording interviews is mentioned, but there is no parallel mention of recording the FGDs and how they were transcribed/translated.

Similarly, informed consent is mentioned for interviews but not FGDs.

Could you further clarify the different roles of the authors in the methodology section? There is mention of a QDMT and analysts as well as data being 'transcribed directly into English locally by one staff member' but whether or not there are overlaps in these roles is unclear.

Could you provide more details about what the 'card statements' are in the methods section? These are mentioned in the table but not described in detail.

2. In the results table, can you clarify if respondents could state if they were living with more than one adult household member, such as both their mother and father, for example? It would be interesting to see how many lived with more than one parent/guardian in terms of risk factors, particularly as there are mentions of losing family members and living with family members who have HIV.

The quotations are lost a little in the table at times and I would have liked to have seen them integrated more into the narrative, but I realise that this may also be due to word count limitations, and that a restructuring is likely not possible at this point!

3. Editing and clarity

The manuscript is well-written, but there are several typos and grammatical errors throughout, particularly in the quotations which would benefit a thorough revision and edit for punctuation, typos, apostrophes and clarity. I recognise the importance of keeping the voice of each individual in a study like this and not editing their language to lose their individuality, but editing for clarity/typos will make their voices more powerful and help the reader be able to fully engage with their words. My comments apply to the quotes throughout, but some specific examples are: ‘anthing’, ‘when she is seeing’, ‘What made to go a ring’ and ‘What made me to leave’.

The word ‘yoh!’ is used in quotes - this appears as 'yho' and 'yoh'.

In the table ‘preferred both equally’ would read better as ‘had no preference’

The side-effect of vaginal fluid is mentioned several times - this is clarified as ‘discharge’ in some of these mentions, but it may be worth explaining more what it means in the quotes as well as reflecting more on the scientific evidence/known side-effects relating to this so that the reader can understand how commonly this is reported and seen in other contexts.

I was interested by the phrasing used on line 429 - is poor persistence the same as poor adherence and/or retention?

Overall, this is a very solid manuscript and the suggestions I make above are minor. Thank you again for the opportunity to review.

6. PLOS authors have the option to publish the peer review history of their article (what does this mean?). If published, this will include your full peer review and any attached files.

Reviewer #1: **Yes: **Tom Decroo

Reviewer #2: No

---

## [Author Response · Author response to Decision Letter 0]

30 Jan 2024

and

RESPONSE: Thank you for this comment. We have checked the manuscript and have ensured that all major sections are set at level 1 heading, are bolded and set at 18pt font. We have further checked the format for levels 2 and 3 headings are set at the correct font. Finally, we have ensured that the rest of the manuscript meets PLOSOne style requirements.

RESPONSE: 

This study was designed and implemented by the Microbicide Trials Network (MTN) funded by the National Institute of Allergy and Infectious Diseases through individual grants (UM1AI068633, UM1AI068615 and UM1AI106707), with cofounding from the Eunice Kennedy Shriver National Institute of Child Health and Human Development and the National Institute of Mental Health, all components of the U.S. National Institutes of Health (NIH)

3. Thank you for stating the following in your Competing Interests section: “The work presented has not been published and is not under consideration in any other peer-reviewed media. The listed authors have all contributed significantly to the design, analysis, and written work, and all authors have given final approval for the version to be published. To the best of our knowledge, no conflict of interest, financial or other, exists.” Please complete your Competing Interests on the online submission form to state any Competing Interests. If you have no competing interests, please state "The authors have declared that no competing interests exist.", as detailed online in our guide for authors at http://journals.plos.org/plosone/s/submit-now

RESPONSE: Thank you. We have ensured that the cover letter contains the above information regarding competing interests.

Additional Editor Comments

The paper offers a comprehensive overview of the advancements made in HIV prevention, focusing on a subject of great interest from the perspective of the beneficiaries. Nonetheless, to increase the manuscript's quality, I believe the authors should heed the suggestions provided by the reviewers. Moreover, I'd like to emphasise the following points:

4. It's crucial for the authors to provide a scientific rationale for the sampling techniques employed and clarify the basis for determining the final sample size. Which specific criteria or references from prior studies influenced this decision?

RESPONSE: We revised the methods section and have included a paragraph stating the rationale of the sampling techniques employed and clarifying the basis for determining the final sample size. Specifically, we cite existing literature and our own previous work in similar studies as having guided the sampling techniques used. We then state that ‘our sample size is advised by ‘information power’ where our recruitment reflected specificity in areas of inquiry (e.g., choosing a product, or choosing neither or those experiencing unique challenges, and, collecting data at specific periods during the study’ as key determinants of the sample size.

5. For clarity, I suggest the results section include a well-organized table detailing the characteristics of the FGDs. This table should outline the number and origin of participants for each FGD, address potential biases due to participant demographics (e.g., all participants of the same age? younger (16) vs older (21)), and specify the FDG locations (e.g., healthcare facility name, county, or neighborhood). Upon examining Table 2, it's challenging to differentiate between FGD and IDI participants.

Response: We have created the table as seen below, but we felt that it did not add much to the findings. We also seem to have many tables and figures, in this paper. So, we appreciate your thoughts on this. Should you review the table below and feel that it is important to add, let us know and we will include in the findings. 

Table 4: Characteristics of FGDs and FGD participants

 FGD Number and site Number of participants per FGD FGD composition by age range 

FGD Composition by product use 

Product adherence category 

Site/Country 

FGD1(DTHF) 3 16, 17, 18 Mixed (Ring, Neither) High Cape Town, South Africa

FGD1(MUJHU) 3 16, 20, 19 Mixed (Ring, Pill) High Kampala, Uganda

FGD1(WRHI) 3 18, 19, 19 Ring High& medium Johannesburg, South Africa

FGD1(UZCHS) 4 16, 16, 18, 19 Ring High Harare, Zimbabwe

FGD2(DTHF) 5 17, 18, 18, 19, 20 Mixed (Ring, Pill, No product) High-low Cape Town, South Africa

FGD2(MUJHU) 5 19, 19, 19, 19, 20 Mixed (Ring, Pill) Mixed Kampala, Uganda

FGD2(WRHI) 4 16, 18, 18, 20 Mixed (Ring, Pill) High-Medium Johannesburg, South Africa

FGD2(UZCHS) 3 17, 17, 18 Mixed (Ring-pill) Medium Harare, Zimbabwe

FGD3(DTHF) 3 19, 19, 21 Mixed (Ring, Pill) High-medium Cape Town, South Africa

FGD3(MUJHU) 4 16, 18, 19, 19, Ring High Kampala, Uganda

FGD3(WRHI) 4 17, 20, 18, 21 Mixed (Ring, pill) High-Medium Johannesburg, South Africa

FGD3(UZCHS) 3 18, 18, 17 Ring High Harare, Zimbabwe

FGD4(DTHF) 5 16, 16, 18, 18, 19 Mixed (No product, Ring) Mixed Cape Town, South Africa

FGD4(MUJHU) 6 18, 18, 19, 19, 20 Ring High-Medium Kampala, Uganda

FGD4(WRHI) 4 16, 19, 19, 19 Mixed (Ring, Pill) High Johannesburg, South Africa

FGD4(UZCHS) 3 17, 17, 18 Mixed (Pill, ring) Medium Harare, Zimbabwe

 Total 

6. To prevent confusion, the term "private place" inside the healthcare facility needs to be defined precisely

RESPONSE: This has been changed in paper. We have changed this from private settings to ‘private rooms’ at the clinical trial site

7. I've noticed the authors refer to "tablet" while interviewees say "pills." The authors should either choose one for uniformity, which is, in my opinion, a better option, or maintain this distinction throughout for consistency. Thank you for your observation, we have ensured that paper refers to ‘Pills’ and not ‘tablet’

RESPONSE: Thank you for your observation, we have ensured that text in the paper refers to ‘Pills’ and not ‘tablet’

8. Based on the findings, the discussion and/or conclusion sections should include more specific recommendations about the implications and continued implementation of HIV programmes in the study sites. Adding more detail to this would improve the paper's clarity and comprehensiveness.

RESPONSE: This is well noted; however, given how different the discussion points are, we opted to include a recommendation/implication under each discussion point as much as possible. We preferred this method to avoid repetition and maintain coherence in the discussion section. We want to draw the reviewer's attention to areas that highlight specific. recommendations/implications under particular discussion points. In lines 1166 to 1180, we share our recommendations based on our discussion point on ‘product triability.’ Lines 1254-1259 share our advice regarding product switching observed during the study and how this is achievable in a real-world setting. Finally, in lines 1262-1268, we provide our thoughts and suggestions regarding what is possible in terms of support strategies that might be most relevant in real-world settings, suggesting that drug-level feedback is not plausible.

Reviewers Comments to the Author

Reviewer #1: This is a very well written paper on a highly relevant topic

Major:

Intro:

9. Please spell out more clearly how the study is novel compared to previous studies. The argument “However, research examining product preference and choice using active products is still lacking” seems to contradict with the previous sentences (“found that discreetness and longer duration of protection to minimize user-burden were favored, along with products that did not interfere with intimate relationships, and provided HIV prevention for unanticipated situations (18). Furthermore, previous experience with a long-acting contraception (e.g., implants) influenced choice of product”) showing factors that inform decision-making

RESPONSE: Thank you for this comment and for the opportunity to clarify. We start by arguing that previous research testing preference used placebo products. Using placebo products means that a participant experiences a product that has no side effects, and yet we know that side effects play a significant role in product choice decisions. To make it clearer, we have stated that ‘research examining product preference and choice using products with active HIV prevention drugs is still lacking, and this is the first of its kind testing preference for oral PrEP containing Emtricitabine/Tenofovir Disoproxil Fumarate and the vaginal ring containing Dapivrine, by offering participants the option to experience both products before choosing their preferred HIV prevention product’.

Methods:

10. What determined the number of FGD and IDI? Was any interim analysis conducted? 

RESPONSE: This comment is well noted, and we have clarified this in point four above. We have added a section titled ‘Qualitative Study Sample’ in which we describe how we determined the sample size (see p6)

11. How did you validate? Was any triangulation (e.g., of data obtained from different stakeholders) done? 

RESPONSE: This comment is well received. As stated in the paper on page 9 of the manuscript, FGD data was our primary data for this paper. Once the analysis was complete and themes developed, we utilized IDI data (which used similar questions) to validate and confirm that data generated on each theme presented information identical to that obtained from FGDs. We also ensured that the development of the FGDs and IDIs guides was robust, having been generated by a team of senior social scientists and researchers at study sites who understand the context better. We further ensured that staff trained in qualitative research and were not part of the staff complement involved in the REACH trial conducted the interviews. All this information is presented in the paper. We have also added a section in the methods section (see page 9) that describes our code development processes, actual coding, and how we measured inter-coder reliability.

12. Table 2: check the formatting of lines. 

RESPONSE: Thank you for this observation. We have ensured that lines are formatted correctly.

Results:

13. A table describing the characteristics of qualitative study participants next to those included in the trial but not in the qualitative study would be relevant.

RESPONSE: We have reviewed the characteristics of the entire REACH sample and compared it to the qualitative choice period sample, and we have yet to find a difference. We suspect that the lack of difference in the characteristics may be partly due to the fact that participants selected for the qualitative also had to reflect the age inclusion criteria, and the mean age of both samples was 19 years. Therefore, such a sample would have similar characteristics around education completion, with few earning an income. The majority of 19-year-olds are not married but have main partners. The majority also chose the ring as their preferred HIV prevention option, as already published elsewhere. We have provided these references in the paper.

Given the information presented in Table 2, adding another column representing the entire REACH sample clutters the table. It overwhelms the reader without adding information different from what is currently shown in this table. Because of this, we have opted to omit to add this information, but should but should the reviewer or editor require this addition, we could add the full sample information as a supplementary table.

14. Emphasize novelty of findings shown, if any. The first paragraph is a good place to do this

RESPONSE: The Novelty of these findings is explained by the fact that they are drawn from participants taking part in the first of its kind, 18-month MTN-034/REACH trial that allowed participants to experience both the ring and the pill for six month and then offered them an opportunity to choose their preferred option during the last six months of the study.. We have now clearly started this in the first paragraph of the discussion section.

15. 441-461: few references – reads like the results section.

 RESPONSE: Additional references in this section have been added

16. 14. Discuss drug level feedback: how could this (or another approach aiming at the same type of info “yes, you ‘re doing well”) work in a programme?

RESPONSE: We note this comment; however, our take-home message is that DLF did not directly inform choice, and, therefore, we find it difficult to make suggestions/ recommendations of an alternative approach that could deliver a similar message if it was not a driving factor. In addition, DLF would be hard and costly to implement in real-world settings. That said, client-centered discussions may be more important than DLF, offering a more subjective experience that participants appreciated more. For example, participants preferred to talk more about getting information on how to engage their families to support their PrEP use., or how to manage side effects

17: Limitations: reflexivity is recommended: how did the experiences and views of the researchers and moderators affect data collection and analysis? Any other bias? Recall bias? Social desirability in a trial on new PREP methods?

RESPONSE: We have added in the limitation section several points about the limitations of qualitative research methods, in general, and how we have addressed both recall, social-desirability and other sources of bias in this research context. (If you want you can add the paragraph once cleaned up that we have at the end of the discussion).

16. Abstract: “use,”: comma should be point

RESPONSE: This has been corrected.

17. Intro: “understand how the use of active products may pose different challenges compared to prior findings in placebo trials” could be “how the comparison of active products” and “placebo-controlled”?

RESPONSE: Thank you for noting this and for the suggestion made. After reviewing the sentence again, we opted to remove that line and have re-written it as follows: “We specifically aimed to understand how experiencing a product first, informed choice, and future roll out of multiple prevention options, using the REACH data”

18. Reviewer #2: Firstly, thank you for the opportunity to review this manuscript. It is very well-written, the research design itself is clear (coming from part of a well-structured, well-known larger study) and the findings are helpful and have an operational relevance to many different contexts. One of the most interesting parts of the study design is the ‘crossover’ period and I think that is something readers can really learn from, so would like to see more made of this in the results.

RESPONSE: We very much appreciate this comment. We have rewritten the section and have made it clearer than before and have cited some of our primary papers that comprehensively describe the cross-over design for the REACH trial. See lines 85-96 on page 33 of the manuscript. For this paper, however, we wanted to focus more on the qualitative data collection methods conducted in the choice period than the trial itself.

19: Can the authors offer a definition of what is meant by 'modern contraceptives'

RESPONSE: When we reviewed and revised the paper, we took this statement and the sentence out entirely because it didn’t provide needed information to the reader and had no impact on the findings presented.

20. This is an interesting study design which clearly a lot of thought and expertise went into - well done! I have made a few comments and suggestion for clarification be

---

## [Decision Letter · Decision Letter 1]

19 Apr 2024

PONE-D-23-24059R1We Choose : Adolescent girls and young women’s choice for an HIV prevention product in a cross-over randomized clinical trial conducted in South Africa, Uganda, and ZimbabwePLOS ONE

Dear Dr. Atujuna,

Thank you for submitting your manuscript to PLOS ONE. After careful consideration, we feel that it has merit but does not fully meet PLOS ONE’s publication criteria as it currently stands. Therefore, we invite you to submit a revised version of the manuscript that addresses the points raised during the review process.

We look forward to receiving your revised manuscript.

Kind regards,

Ivan Alejandro Pulido Tarquino, MSc

Academic Editor

PLOS ONE

Additional Editor Comments:

Dera Authors,

please address the comments provided by one of the reviewers during his second review.

Thank you

Kind regards

Reviewers' comments:

Reviewer's Responses to Questions

**Comments to the Author**

1. If the authors have adequately addressed your comments raised in a previous round of review and you feel that this manuscript is now acceptable for publication, you may indicate that here to bypass the “Comments to the Author” section, enter your conflict of interest statement in the “Confidential to Editor” section, and submit your "Accept" recommendation.

Reviewer #1: (No Response)

Reviewer #2: All comments have been addressed

2. Is the manuscript technically sound, and do the data support the conclusions?

Reviewer #1: Yes

Reviewer #2: (No Response)

3. Has the statistical analysis been performed appropriately and rigorously? 

Reviewer #1: N/A

Reviewer #2: (No Response)

4. Have the authors made all data underlying the findings in their manuscript fully available?

Reviewer #1: No

Reviewer #2: (No Response)

5. Is the manuscript presented in an intelligible fashion and written in standard English?

Reviewer #1: Yes

Reviewer #2: (No Response)

6. Review Comments to the Author

Reviewer #1: Dear authors

The arguments shown in the rebuttal and the modifications that you propose are satisfactory.

However ....

With regards to point 9. You wrote in the rebuttal that you added in the manuscript text "To make it clearer, we have stated that ‘research examining product preference and choice using products with active HIV prevention drugs is still lacking, and this is the first of its kind testing preference for oral PrEP containing Emtricitabine/Tenofovir Disoproxil Fumarate and the vaginal ring containing Dapivrine, by offering participants the option to experience both products before choosing their preferred HIV prevention product’

However, this info was not shown in the revised version of the manuscript.

Please revised the manuscript and ensure that it corresponds with what is mentioned in the rebuttal.

It is not feasible for me to check whether the revised manuscript corresponds with what you mention in the rebuttal as modification.

Point 10 in your rebuttal: "What determined the number of FGD and IDI? Was any interim analysis conducted?". You refer to point 4. Thiank you for the info. You refer to information power. How did you assess whether you had collected enough information, and that additional data collection would not result in additional new insights (thus whether saturation was reached)?

Reviewer #2: (No Response)

7. PLOS authors have the option to publish the peer review history of their article (what does this mean?). If published, this will include your full peer review and any attached files.

Reviewer #1: **Yes: **Tom Decroo

Reviewer #2: No

---

## [Author Response · Author response to Decision Letter 1]

3 Jun 2024

4: Data availability: We have noted the PLOS data policy which requires authors to make all data underlying the findings described in their manuscript fully available without restriction, with rare exception (please refer to the Data Availability Statement in the manuscript PDF file). Due to the composition of the study sites with varying confidentiality agreements, our response is as follows: Study data are available upon request from the Microbicide Trials Network by submission of a “Dataset Request Form” available at http://www.mtnstopshiv.org/resources. Interested parties would be able to access these data in the same manner as the authors. The authors did not have any special access privileges that others would not have.”

Our response to the specific reviewers' comments is outlined below:

Reviewer #1: Dear authors, the arguments shown in the rebuttal and the modifications that you propose are satisfactory. However, with regards to point 9. You wrote in the rebuttal that you added in the manuscript text "To make it clearer, we have stated that ‘research examining product preference and choice using products with active HIV prevention drugs is still lacking, and this is the first of its kind testing preference for oral PrEP containing Emtricitabine/Tenofovir Disoproxil Fumarate and the vaginal ring containing Dapivrine, by offering participants the option to experience both products before choosing their preferred HIV prevention product’ However, this info was not shown in the revised version of the manuscript. Please revised the manuscript and ensure that it corresponds with what is mentioned in the rebuttal. It is not feasible for me to check whether the revised manuscript corresponds with what you mention in the rebuttal as modification.

RESPONSE: We apologise for the oversight and thank you for bringing this to our attention. We have included the statement…‘research examining product preference and choice using products with active HIV prevention drugs is still lacking, and this is the first of its kind testing preference for oral PrEP containing Emtricitabine/Tenofovir Disoproxil Fumarate and the vaginal ring containing Dapivrine, by offering participants the option to experience both products before choosing their preferred HIV prevention product’, on page 3 of the paper.

Reviewer #1 comment: Point 10 in your rebuttal: "What determined the number of FGD and IDI? Was any interim analysis conducted?". You refer to point 4. Thank you for the info. You refer to information power. How did you assess whether you had collected enough information, and that additional data collection would not result in additional new insights (thus whether saturation was reached)?

RESPONSE: Correct, we conducted interim analysis using what we call debriefing reports. We thus have added a statement that…‘Additionally, during data collection, we promptly analysed debriefing reports (DRs) from completed interviews before the conclusion of data collection. These debriefing reports synthesized the main themes emerging from each interview, with interviewers completing them within 4-7 days of conducting the interview. This rapid analysis enabled us to ascertain data saturation. This is added on page 33.

---

## [Decision Letter · Decision Letter 2]

26 Jul 2024

We Choose : Adolescent girls and young women’s choice for an HIV prevention product in a cross-over randomized clinical trial conducted in South Africa, Uganda, and Zimbabwe

PONE-D-23-24059R2

Dear Dr. Millicent Atujuna,

We’re pleased to inform you that your manuscript has been judged scientifically suitable for publication and will be formally accepted for publication once it meets all outstanding technical requirements.

Kind regards,

Ivan Alejandro Pulido Tarquino, MSc

Academic Editor

PLOS ONE

Additional Editor Comments (optional):

Dear Author(s),

I am pleased to inform you that your manuscript has been accepted for publication in our journal. I would like to highlight several strengths of your work that make a significant contribution to the field and should encourage you to continue your important research on this crucial topic.

Firstly, your study's multi-country perspective within a region of Africa with a notably high prevalence of HIV is commendable. This approach underscores the necessity of addressing the HIV epidemic among those at highest risk, providing valuable insights that are relevant across multiple contexts. Secondly, the factors motivating participants to engage in your study are particularly noteworthy. By highlighting these motivations, your work sets a valuable example for similar studies in the region, demonstrating effective strategies for participant recruitment and engagement. Lastly, I find it extremely important that your study captured the perspectives of the beneficiaries regarding the PrEP products and their characteristics. Your emphasis on the various available options and the strong preference for less frequent dosing—embodied in the "Set it and forget it" —provides critical insights into user preferences that can guide future interventions and product development.

Once again, congratulations on your excellent work. We look forward to your future contributions to this vital area of study.

Best regards

Reviewers' comments:

Reviewer's Responses to Questions

**Comments to the Author**

1. If the authors have adequately addressed your comments raised in a previous round of review and you feel that this manuscript is now acceptable for publication, you may indicate that here to bypass the “Comments to the Author” section, enter your conflict of interest statement in the “Confidential to Editor” section, and submit your "Accept" recommendation.

Reviewer #1: All comments have been addressed

2. Is the manuscript technically sound, and do the data support the conclusions?

Reviewer #1: Yes

3. Has the statistical analysis been performed appropriately and rigorously? 

Reviewer #1: N/A

4. Have the authors made all data underlying the findings in their manuscript fully available?

Reviewer #1: No

5. Is the manuscript presented in an intelligible fashion and written in standard English?

Reviewer #1: Yes

6. Review Comments to the Author

Reviewer #1: Dear authors

Thank you for addressing my comments.

In my view the paper is ready to be accepted for publication.

best wishes

7. PLOS authors have the option to publish the peer review history of their article (what does this mean?). If published, this will include your full peer review and any attached files.

Reviewer #1: **Yes: **Tom Decroo

---

## [Editor Report · Acceptance letter]

20 Aug 2024

PONE-D-23-24059R2 

PLOS ONE

Dear Dr. Atujuna, 

I'm pleased to inform you that your manuscript has been deemed suitable for publication in PLOS ONE. Congratulations! Your manuscript is now being handed over to our production team.

Kind regards, 

on behalf of

Dr. Ivan Alejandro Pulido Tarquino 

Academic Editor

PLOS ONE